# Initiations of safer supply hydromorphone increased during the COVID-19 pandemic in Ontario: An interrupted time series analysis

Samantha Young[1,2,3]*, Tara Gomes[1,4,5,6], Gillian Kolla[7], Daniel McCormack[5], Zoë Dodd[4], Janet Raboud[8], Ahmed M. Bayoumi[1,4,5,9]

**1** Institute of Health Policy, Management and Evaluation, University of Toronto, Toronto, Ontario, Canada, **2** Interdepartmental Division of Addiction Medicine, St. Paul's Hospital, Vancouver, British Columbia, Canada, **3** British Columbia Centre on Substance Use, Vancouver, British Columbia, Canada, **4** MAP Centre for Urban Health Solutions, St. Michael's Hospital, Toronto, Ontario, Canada, **5** ICES, Toronto, Ontario, Canada, **6** Leslie Dan Faculty of Pharmacy, University of Toronto, Toronto, Ontario, Canada, **7** Canadian Institute for Substance Use Research, University of Victoria, Victoria, British Columbia, Canada, **8** Dalla Lana School of Public Health, University of Toronto, Toronto, Ontario, Canada, **9** Division of General Internal Medicine, Department of Medicine, St. Michael's Hospital, Unity Health, Toronto, Ontario, Canada

* youngsamanthav@gmail.com

**Data Availability Statement:** The dataset from this study is held securely in coded form at the Institute for Clinical Evaluative Sciences (ICES). While data sharing agreements prohibit ICES from publicly

## Abstract

### Aims

Calls to prescribe safer supply hydromorphone (SSHM) as an alternative to the toxic drug supply increased during the COVID-19 pandemic but it is unknown whether prescribing behaviour was altered. We aimed to evaluate how the number of new SSHM dispensations changed during the pandemic in Ontario.

### Methods

We conducted a retrospective interrupted time-series analysis using provincial administrative databases. We counted new SSHM dispensations in successive 28-day periods from March 22, 2016 to August 30, 2021. We used segmented Poisson regression methods to test for both a change in level and trend of new dispensations before and after March 17, 2020, the date Ontario's pandemic-related emergency was declared. We adjusted the models to account for seasonality and assessed for over-dispersion and residual autocorrelation. We used counterfactual analysis methods to estimate the number of new dispensations attributable to the pandemic.

### Results

We identified 1489 new SSHM dispensations during the study period (434 [mean of 8 per 28-day period] before and 1055 [mean of 56 per 28-day period] during the pandemic). Median age of individuals initiating SSHM was 40 (interquartile interval 33–48) with 61.7% (N = 919) male sex. Before the pandemic, there was a small trend of increased prescribing (incidence rate ratio [IRR] per period 1.002; 95% confidence interval [95CI] 1.001–1.002; p<0.001), with a change in level (immediate increase) at the pandemic date (relative

releasing a minimal deidentified dataset, access can be granted to those who meet pre-specified criteria for confidential access through the Data & Analytics Service (DAS). More information is available at www.ices.on.ca/DAS [email: das@ices.on.ca]). Instructions for submitting a data request can be found at the following link: https://www.ices.on.ca/DAS/Submitting-your-request.

**Funding:** This project was supported by grant funding from the Canadian Institutes of Health Research (CIHR; https://cihr-irsc.gc.ca/e/193.html) (grant # 448935) and the Ontario Ministry of Health (https://www.health.gov.on.ca/en/) (grant #709). This study was supported by ICES (https://www.ices.on.ca/), which is funded by an annual grant from the Ontario Ministry of Health (MOH) and the Ministry of Long-Term Care (MLTC). This work was also supported by the Ontario Health Data Platform (OHDP), a Province of Ontario initiative to support Ontario's ongoing response to COVID-19 and its related impacts. Parts of this material are based on data and information compiled and provided by the Ontario Ministry of Health, Ontario Health (OH) and CIHI. This document used data adapted from the Statistics Canada Postal CodeOM Conversion File, which is based on data licensed from Canada Post Corporation, and/or data adapted from the Ontario Ministry of Health Postal Code Conversion File, which contains data copied under license from ©Canada Post Corporation and Statistics Canada. We thank IQVIA Solutions Canada Inc. for use of their Drug Information File. The analyses, conclusions, opinions and statements expressed herein are solely those of the authors and do not reflect those of the funding or data sources; no endorsement is intended or should be inferred. Samantha Young is supported by the CIHR Vanier Canada Graduate Scholarships and the International Collaborative Addiction Medicine Research Fellowship (NIDA grant R25-DA037756). Gillian Kolla is supported by a Canadian Network on Hepatitis C (CanHepC) Postdoctoral Fellowship and a Canadian Institutes of Health Research Banting Postdoctoral Researcher Award. Tara Gomes is supported by a Tier 2 Canada Research Chair. Ahmed Bayoumi is supported by the Baxter and Alma Ricard Chair in Inner City Health at St. Michael's Hospital and the University of Toronto. The funders had no role in study design, data collection and analysis, decision to publish, or preparation of the manuscript.

**Competing interests:** The authors have declared that no competing interests exist.

increase in IRR 1.674; 95CI 1.206–2.322; p = 0.002). The trend during the pandemic was not statistically significant (relative increase in IRR 1.000; 95CI 1.000–1.001; p = 0.251). We estimated 511 (95CI 327–695) new dispensations would not have occurred without the pandemic.

## Conclusion

The pandemic led to an abrupt increase in SSHM prescribing in Ontario, although the rate of increase was similar before and during the pandemic. The absolute number of individuals who accessed SSHM remained low throughout the pandemic.

## Introduction

The North American unregulated opioid drug supply is characterized by high potency synthetic opioids–primarily illicitly manufactured fentanyl–often adulterated with other drugs that increase the risk of overdose and overdose death [1, 2]. One response to the resulting unregulated drug toxicity crisis is often referred to as "safer supply," whereby people who use drugs are prescribed pharmaceutical opioids to reduce their reliance on the unregulated drug supply [3–5]. This harm-reduction based prescribing practice is also based on the recognition that some individuals do not find opioid agonist therapy, including methadone and buprenorphine, to be effective in meeting their goals [3].

Some physicians and nurse practitioners in Canada started prescribing safer supply hydromorphone (SSHM) as early as 2016 [6, 7]. From January 2016 to March 2020, SSHM prescribing increased in Ontario, although fewer than 450 individuals received safer supply [7]. In the first year of the pandemic, opioid toxicity-related deaths increased by 95% in Canada and overdose-related deaths from any substance increased by up to 60% in some jurisdictions in the United States [1, 8]. While the causes of increased overdose-related mortality during the pandemic are likely multifactorial, the persistent volatility of the unregulated drug supply is suspected to be a contributing factor [9].

Some Canadian policymakers, physicians, researchers, and people who use drugs viewed SSHM as a public health strategy to curb overdose deaths and promote physical distancing [10–13]. Shortly after the pandemic began, the Canadian government announced increased funding for both new programs and expansion of existing safer supply initiatives including the expansion of several community-based safer opioid supply programs in Ontario cities including London, Toronto, Ottawa, and Hamilton [10]. Additionally, several Canadian COVID-19 recovery and isolation hotels for people experiencing homelessness, including some in Ontario, offered SSHM in addition to opioid agonist therapy [14, 15]. In British Columbia, guidelines were released in conjunction with the provincial government to support the prescribing of SSHM as a response to the pandemic [13]. In Ontario, the pre-existing safer supply guidance document was updated for the pandemic context [16]. Nevertheless, the prescribing of SSHM remained controversial due to the unwitnessed nature of most SSHM use, the potential risk of infection, concerns about diversion, and the lack of robust evidence on its effectiveness [17]. We hypothesized that the prescribing and guideline changes related to the declaration of a pandemic emergency altered SSHM prescribing and aimed to evaluate this through examining how the number of initiations of SSHM changed before and after the declaration of an emergency due to the COVID-19 pandemic in Ontario.

## Methods

### Overview

We conducted a retrospective interrupted time-series analysis of changes in prescribing of SSHM after the declaration of the COVID-19 pandemic [18]. We used Ontario administrative databases held at ICES (formerly the Institute for Clinical Evaluative Sciences). We used the Narcotic Monitoring System database to identify hydromorphone and opioid agonist therapy prescribing; this database records information for all controlled substances (including prescription opioids) dispensed from community pharmacies in Ontario, regardless of payer. We also used the Canadian Institute for Health Information's (CIHI) Discharge Abstract Database, the National Ambulatory Care Reporting System, the Ontario Health Insurance Plan (OHIP) database, and the OHIP Registered Persons Database to identify diagnoses and procedures during inpatient hospital admissions and emergency department visits, and to describe demographic characteristics of the cohort (S2 Table). These datasets were linked using unique encoded identifiers and analyzed at ICES. ICES is an independent, non-profit research institute whose legal status under Ontario's health information privacy law allows it to collect and analyze health care and demographic data, without consent, for health system evaluation and improvement. ICES is a prescribed entity under Ontario's Personal Health Information Protection Act (PHIPA). Section 45 of PHIPA authorizes ICES to collect personal health information, without consent, for the purpose of analysis or compiling statistical information with respect to the management of, evaluation or monitoring of, the allocation of resources to or planning for all or part of the health system. Projects that use data collected by ICES under section 45 of PHIPA, and use no other data, are exempt from REB review. The use of the data in this project is authorized under section 45 and approved by ICES' Privacy and Legal Office. All analyses were completed on de-identified data. We followed the Strengthening the Reporting of Observational Studies in Epidemiology (STROBE) guidelines for observational studies.

### Eligibility and classification

We included individuals with opioid use disorder who received immediate release hydromorphone doses consistent with SSHM prescribing between March 22, 2016 and August 30, 2021. We chose to begin the study period in 2016 because it is when SSHM prescribing first occurred in Ontario. We defined the pre-pandemic and pandemic periods based on dates occurring before versus on or after March 17, 2020, the date a pandemic-related state of emergency was declared in Ontario, respectively [18]. We defined safer supply hydromorphone prescribing based on a definition we developed for a prior study [7]. Specifically, we classified hydromorphone prescribing as consistent with SSHM if the following criteria were met: 1) patients were dispensed doses totaling at least 32 mg per day (using 4 mg or 8 mg oral formulations) for at least 2 of 3 consecutive days; and 2) prescriptions were dispensed daily. We set the index date as the first date of SSHM prescribing. We classified individuals as having opioid use disorder if they met at least one of the following criteria: 1) used opioid use disorder-specific health services in the 2 years prior to the index date; 2) prescription for methadone or buprenorphine in the 4 years before the index date; and 3) opioid toxicity treated in an emergency department or hospital within 2 years before the index date (S1 Table). Individuals were excluded if they were not Ontario residents, had invalid or missing age or sex data, or died on or before the first day of the first study period. We also excluded individuals who had a cancer diagnosis or cancer-related treatment in the 1 year before the index date (S1 Table) because of the potential for hydromorphone prescribed for pain to be misclassified as SSHM. Each individual contributed

only one dispensation to the study and were included at their first instance of meeting the cohort definition chronologically during the study period.

## Analysis

We summarized baseline sociodemographic and health-related characteristics of individuals prescribed SSHM using descriptive statistics. Individuals initiating safer supply hydromorphone in the pre- versus pandemic periods were compared using Pearson's chi-squared test for binary measures and Wilcoxon rank sum test for continuous measures.

We counted new dispensations of SSHM in Ontario in successive 28-day periods over the study period. In addition to analyzing new dispensations overall, we performed subgroup analyses by male and female sex (based on sex documented at birth). In periods with 1 to 5 dispensations inclusively, the number was suppressed in accordance with ICES privacy policies. We assigned these months a random integer value within this range with equal probability.

We analyzed these data as an interrupted time series of counts using segmented regression methods and tested for both a change in level and a change in trend related to the COVID-19 pandemic [19, 20].

We addressed three considerations in specifying the regression model. First, we assessed the model for seasonality after visual inspection of trends over time suggested periodic trends in the data. We entered Fourier term coefficients into the model, which allow for modeling period fluctuations as harmonic (sine and cosine) terms [21]. We decided on the number of harmonics by testing model fit after entering one to three pairs of terms and assessing fit using Bayesian Information Criteria (BIC). We used a monthly frequency based on visual inspection of trends over time, a periodogram, and clinical judgment [22]. Second, we assessed two models for over-dispersed count data using the BIC: a Poisson model with robust variance estimators and a negative binomial model [23]. Third, we assessed models for residual autocorrelation through visual inspection of autocorrelation and partial autocorrelation plots [20]. We evaluated overall model fit through both visual inspection of graphs comparing observed to fitted values and graphs of Pearson residuals over time.

We used counterfactual analysis methods to estimate the number of new dispensations attributable to the pandemic. To do so, we used predictive margins to estimate the number of new dispensations in each period within our analysis after March 17, 2020, both with and without pandemic effects and used the delta method to estimate uncertainty bounds around the difference [24, 25].

We conducted three sensitivity analyses. First, we assessed the sensitivity of our results to the imputation of censored values for months with few dispensations by examining the stability of regression parameter estimates for pre-pandemic trends, the level change at the time the pandemic was called, and the pandemic trend over 1000 simulations. We also estimated the number of new dispensations attributable to the pandemic in each of these samples. Second, we selected a change date based on visual inspection of dispensation rates over time, which suggested that the change may have occurred later than the date the pandemic emergency was declared. A later date is consistent with the concept that alterations in dispensation, such as changed prescribing habits, may have been implemented some weeks after the declaration of emergency. Third, we examined an alternative definition for eligibility criteria. Our primary analysis included individuals who were dispensed immediate release hydromorphone prior to their index date because it is known that people with opioid use disorder are sometimes prescribed short acting opioids outside of safer supply prescribing before starting SSHM [7, 26]. In this sensitivity analysis, we excluded individuals who were dispensed 4 or 8 mg tablets of

hydromorphone in the 14 days before the index date, which we anticipate was a more specific (but less sensitive) algorithm.

All reported p-values are two-sided with a type I error rate of 5%.

## Results

### Primary analysis

We identified 1489 initiations of SSHM during the study period, of which 434 (8 per 28-day period) were before and 1055 (56 per 28-day period) after the declaration of the pandemic. The median age was 40 (interquartile range [IQR] 33–48), and individuals initiated on SSHM prior to the pandemic were older than those initiated during the pandemic (median age 42 [IQR 34–50] versus median age 39 [IQR 33–47], respectively; p = <0.001) (Table 1). Overall, most individuals (61.7%; N = 919) were male and more than half (52.9%; N = 788) were in the lowest income quintile, both of which did not differ significantly between the two periods. Individuals initiated on SSHM during the pandemic, compared with individuals initiated before the pandemic, were less likely to have a diagnosis of HIV (5.5% versus 14.1%, p = <0.001); have had an infective complication from infective endocarditis, osteomyelitis, discitis, or skin and soft tissue infection (13.6% versus 26.5%, p = <0.001) (S1 Table); or have received palliative care services in the prior 180 days (≤0.3% versus 13.4%, p = <0.001) (S1 Table); and were more likely to have had an opioid-related overdose in the prior 1 year (29.6% versus 14.1%, p = <0.001) (S1 Table); have been hospitalized for a substance-related disorder in the prior 3 years (52.9% versus 42.2%, p<0.001); or have been prescribed any opioid agonist therapy in the prior one year (82.6% versus 67.7%, p = <0.001).

The number of new SSHM dispensations per 28-day period remained low until October 2018 (≤10), after which it began to increase slightly. The peak number of dispensations per 28-day period prior to the pandemic occurred in September, 2019 at 32. The rate increased further after the declaration of the pandemic in March 2020 (Fig 1). The peak number of new dispensations occurred between August 4 and November 23, 2020, with an average of 83 initiations per 28-day period. The best-fitting model was a Poisson regression which included 2 pairs of Fourier terms; this model had no residual autocorrelation and acceptable model fit. Prior to the pandemic, there was a small increase in rate of new SSHM (incidence rate ratio [IRR] 1.002 per 28-day period; 95% confidence interval [95CI] 1.001 to 1.002; p<0.001) (Table 2); this corresponds to an increase of about 2.6% ($1.002^{365.25/28}$) per year. There was an immediate increase in new SSHM dispensations at the time the pandemic was declared (relative increase in IRR 1.674; 95CI 1.206 to 2.322; p = 0.002); however, the trend during the pandemic was not statistically significantly different than the trend prior to the pandemic (relative increase in IRR per period of 1.000; 95CI 1.000 to 1.001; p = 0.251). We estimated that there were 511 (95CI 327 to 695) new SSHM dispensations (27 per 28-day period) attributable to the pandemic in the 532 days after the pandemic was declared.

### Sex-specific analyses

When stratified by sex, 570 females (180 [3 per 28-day period] before and 390 [20 per 28-day period] during the pandemic) and 919 males (254 [5 per 28-day period] before and 665 [35 per 28-day period] during the pandemic) were initiated on SSHM. In a subgroup analysis by sex, the best fitting model was a Poisson model with 2 and 1 Fourier terms for males and females, respectively. The pre-pandemic trend for both sexes showed a similar small increase in prescribing per period (IRR 1.002 for both per period; 95CI 1.001 to 1.002 for males and 1.001 to 1.003 for females; p<0.001 for both). The pandemic-related level change in rate was significant for males (relative increase in IRR 1.691; 95CI 1.075 to 2.658; p = 0.023) but not for females (relative increase in IRR 1.289; 95CI 0.912 to 1.821; p = 0.15) (Fig 2). The pandemic trend was

**Table 1. Baseline characteristics of patients initiating safer supply immediate release hydromorphone, stratified by the pre-pandemic (March 22, 2016 –March 16, 2020) and pandemic (March 17, 2020 –August 30, 2021) periods.**

| Characteristic | All Initiations | Pre-pandemic Initiations | Pandemic Initiations | P-value |
|---|---|---|---|---|
| | n (%) | n (%) | n (%) | |
| | n = 1489 | n = 434 | n = 1055 | |
| *Sociodemographic Characteristics* | | | | |
| **Age (in years)** | | | | |
| Median (IQR) | 40 (33–48) | 42 (34–50) | 39 (33–47) | <0.001 |
| **Sex** | | | | |
| Male | 919 (61.7) | 254 (58.5) | 665 (63.0) | 0.10 |
| Female | 570 (38.3) | 180 (41.5) | 390 (37.0) | |
| **Location of Residence** | | | | |
| Urban | 1411 (94.8) | 420 (96.8) | 991 (93.9) | 0.065 |
| Rural | 37 (2.5) | 8 (1.8) | 29 (2.7) | |
| Missing | 41 (2.8) | 6 (1.4) | 35 (3.3) | |
| **Income Quintile** | | | | 0.087 |
| 1 (lowest) | 788 (52.9) | 249 (57.4) | 539 (51.1) | |
| 2 | 295 (19.8) | 81 (18.7) | 214 (20.3) | |
| 3 | 177 (11.9) | 54 (12.4) | 123 (11.7) | |
| 4 | 101 (6.8) | 22 (5.1) | 79 (7.5) | |
| 5 (highest) | 86 (5.8) | 21 (4.8) | 65 (6.2) | |
| Missing | 42 (2.8) | 7 (1.6) | 35 (3.3) | |
| *Health-related characteristics* | | | | |
| **HIV seropositive prior to initiation** | 119 (8.0) | 61 (14.1) | 58 (5.5) | <0.001 |
| **Infective complication in prior 1 year** | | | | |
| Any | 259 (17.4) | 115 (26.5) | 144 (13.6) | <0.001 |
| Infective endocarditis | 36 (2.4) | 17 (3.9) | 19 (1.8) | 0.016 |
| Osteomyelitis, discitis, or SSTI | 247 (16.6) | 112 (25.8) | 135 (12.8) | <0.001 |
| **Received palliative care services in prior 180 days** | 61 (4.1) | 56–61 (12.9–14.1) | ≤5 (≤0.5) | <0.001 |
| **Opioid-related overdose in prior 1 year** | 373 (25.1) | 61 (14.1) | 312 (29.6) | <0.001 |
| **ED visit or hospitalization in prior 3 years for:** | | | | |
| Substance-related disorder | 741 (49.8) | 183 (42.2) | 558 (52.9) | <0.001 |
| Deliberate self-harm | 308 (20.7) | 84 (19.4) | 224 (21.2) | 0.42 |
| *Medication characteristics* | | | | |
| **Benzodiazepines in prior 30 days** | 209 (14.0) | 72 (16.6) | 137 (13.0) | 0.069 |
| **Methadone in prior 1 year** | 937 (62.9) | 235 (54.1) | 702 (66.5) | <0.001 |
| **Buprenorphine/naloxone in prior 1 year** | 361 (24.2) | 104 (24.0) | 257 (24.4) | 0.87 |
| **Daily dispensed slow release oral morphine in prior 1 year** | 370 (24.8) | 47 (10.8) | 323 (30.6) | <0.001 |
| **Any opioid agonist therapy in prior 1 year** | 1165 (78.2) | 294 (67.7) | 871 (82.6) | <0.001 |

IQR = interquartile range; SSTI = skin or soft tissue infection. ED = emergency department. Opioid agonist therapy includes methadone, buprenorphine/naloxone, and daily dispensed slow release oral morphine.

not statistically different from the pre-pandemic trend for either sex (p = 0.66 for males, p = 0.90 for females, Table 2).

## Sensitivity analyses

**Imputed low numbers.** Over 1000 simulations that used imputed values for 24 censored values with few (<5) new dispensations, the regression parameter estimates for pre-pandemic

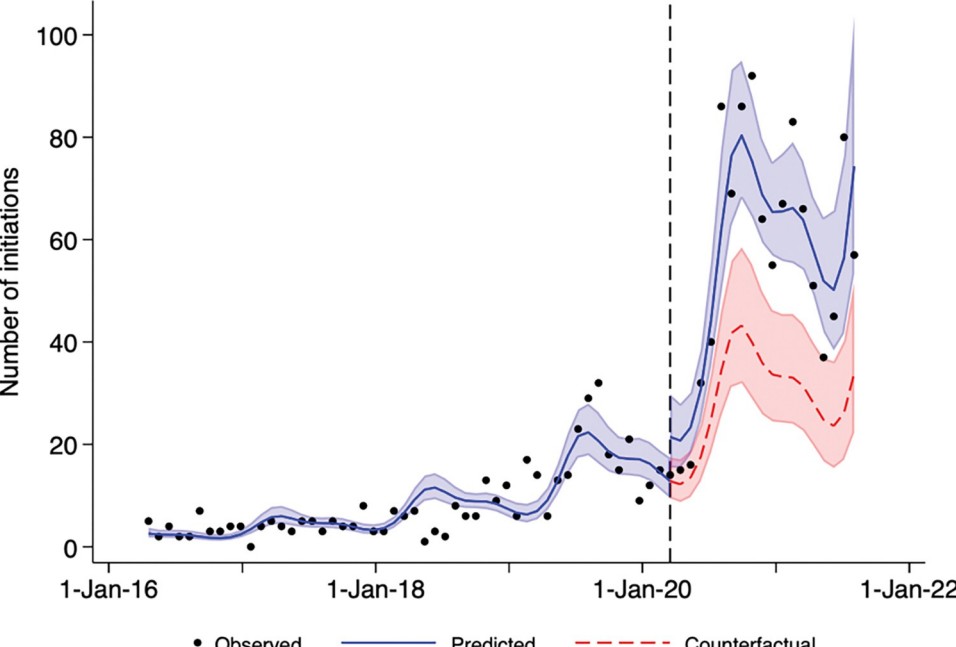

**Fig 1. New safer supply hydromorphone dispensations over time in Ontario.** Dots represent the number of safer supply hydromorphone dispensations per 28-day period between early 2016 and late 2021 in Ontario. The vertical dashed line indicates March 17, 2020, the date that Ontario declared an emergency due to the COIVD-19 pandemic. The solid blue line indicates predictions from the best-fitting Poisson regression model and light blue shading indicates confidence bounds around these estimates. The dashed red line indicates the estimated number of dispensations if the pre-pandemic trends had continued during the pandemic (i.e., the counterfactual scenario) and light red shading indicates confidence bounds around these estimates.

trends, the level change, and the pandemic trend were stable; the first two parameters were statistically significant at the p<0.05 threshold in all simulations and at the p = 0.001 threshold in 83.2% of simulations. The parameter for the trend during the pandemic was not significant (p>0.05) in any simulation. The low and high estimates of the number of new SSHM dispensations attributable to the pandemic were 347 (95CI 92 to 602) and 537 (95CI 352 to 723), respectively.

**Alternate change date.**  Visual inspection suggested that changes in dispensation occurred approximately 2 periods (56 days) after the declaration of the pandemic. Using this date (May 12, 2022) as the change date yielded similar regression coefficients (Table 2) and a slightly improved fit over the baseline model (BIC 472.2 for the alternate date model vs. 480.1 for the baseline model). In these analyses, the models estimated 497 (95CI 333 to 661) new SSHM dispensations attributable to the pandemic over the 476 days after the change date (17 new dispensations per 28-day period).

**Excluding individuals recently dispensed hydromorphone.**  Excluding individuals who had been dispensed 4mg or 8mg hydromorphone tablets in the 14 days prior to the index date resulted in 402 fewer initiations of SSHM for a total of 1087 initiations during the study period (241 before and 846 after the declaration of the pandemic). The regression coefficients were similar to the baseline model (Table 2) with a slightly higher relative increase in initiations at the onset of the pandemic compared with the primary analysis (IRR 1.783; 95CI 1.335 to 2.383; p<0.001).

**Table 2. Primary, sex-specific, and sensitivity analyses comparing initiations of safer supply hydromorphone before and during the COVID-19 pandemic.**

| | Incidence Rate Ratio | (95% Confidence Interval) | P-value |
|---|---|---|---|
| **Baseline model–All individuals** | | | |
| Pre-pandemic trend | 1.002 | (1.001 to 1.002) | <0.001 |
| Pandemic level change | 1.674 | (1.206 to 2.322) | 0.002 |
| Pandemic trend | 1.001 | (1.000 to 1.001) | 0.25 |
| **Sex-specific model–Males** | | | |
| Pre-pandemic trend | 1.002 | (1.001 to 1.002) | <0.001 |
| Pandemic level change | 1.691 | (1.075 to 2.658) | 0.023 |
| Pandemic trend | 1.000 | (0.999 to 1.002) | 0.66 |
| **Sex-specific model–Females** | | | |
| Pre-pandemic trend | 1.002 | (1.001 to 1.003) | <0.001 |
| Pandemic level change | 1.289 | (0.912 to 1.821) | 0.15 |
| Pandemic trend | 1.000 | (0.999 to 1.001) | 0.90 |
| **Sensitivity analysis excluding individuals who received 4mg or 8mg hydromorphone in 14 days prior to index date** | | | |
| Pre-pandemic trend | 1.002 | (1.002 to 1.002) | <0.001 |
| Pandemic level change | 1.783 | (1.335 to 2.383) | <0.001 |
| Pandemic trend | 1.001 | (1.000 to 1.001) | 0.25 |
| **Sensitivity analysis with alternate intervention date May 12, 2020** | | | |
| Pre-pandemic trend | 1.002 | (1.001 to 1.002) | <0.001 |
| Pandemic level change | 1.925 | (1.381 to 2.685) | <0.001 |
| Pandemic trend | 1.000 | (0.999 to 1.001) | 0.955 |

Parameters for Fourier terms and the model constant can be found in S2 Table. All models use an intervention date of March 17, 2020 unless otherwise specified. Pre-pandemic trend refers to the trend from March 22, 2016 to March 16, 2020. Pandemic level change refers to the change at the onset of the pandemic on March 17, 2020. Pandemic trend refers to the trend from March 17, 2020 to August 30, 2021.

## Discussion

Our study used provincial administrative health data from more than five years to provide a comprehensive population-level overview of SSHM initiations in Ontario. We found that rates of prescribing of SSHM were increasing prior to the pandemic. Prescribing increased significantly and immediately after the COVID-19 public health emergency was declared in Ontario. The ratesd of prescribing during the pandemic were explained by a combination of both the pre-pandemic trend and the immediate increase. That is, we did not observe an additional increase in prescribing rates during the pandemic beyond this combination.

We found that rates of SSHM prescribing were low but increasing slowly prior to the pandemic; we suspect that this was related to increased funding for programs prescribing safer supply and advocacy by SSHM prescribers and community members to expand access [7]. Our results indicate that the onset of the pandemic prompted an abrupt increase in SSHM prescribing within two months of the declaration of the public health emergency, and we estimate that approximately half of the initiations would not have occurred if there had not been a pandemic. Visual inspection of the time trends and a sensitivity analysis suggested a delay between the declaration of the public health emergency and increased uptake in SSHM prescribing, which likely reflects the lag in development and dissemination of information and resources. A recent national environmental scan identified several important factors contributing to the reported expansion of SSHM during the pandemic including the release of rapidly-developed or revised provincial and local guidance documents supporting SSHM prescribing, expansion of funding for safer supply initiatives, and concerns regarding the heightened risk of COVID-19 and overdose among people who use opioids [12].

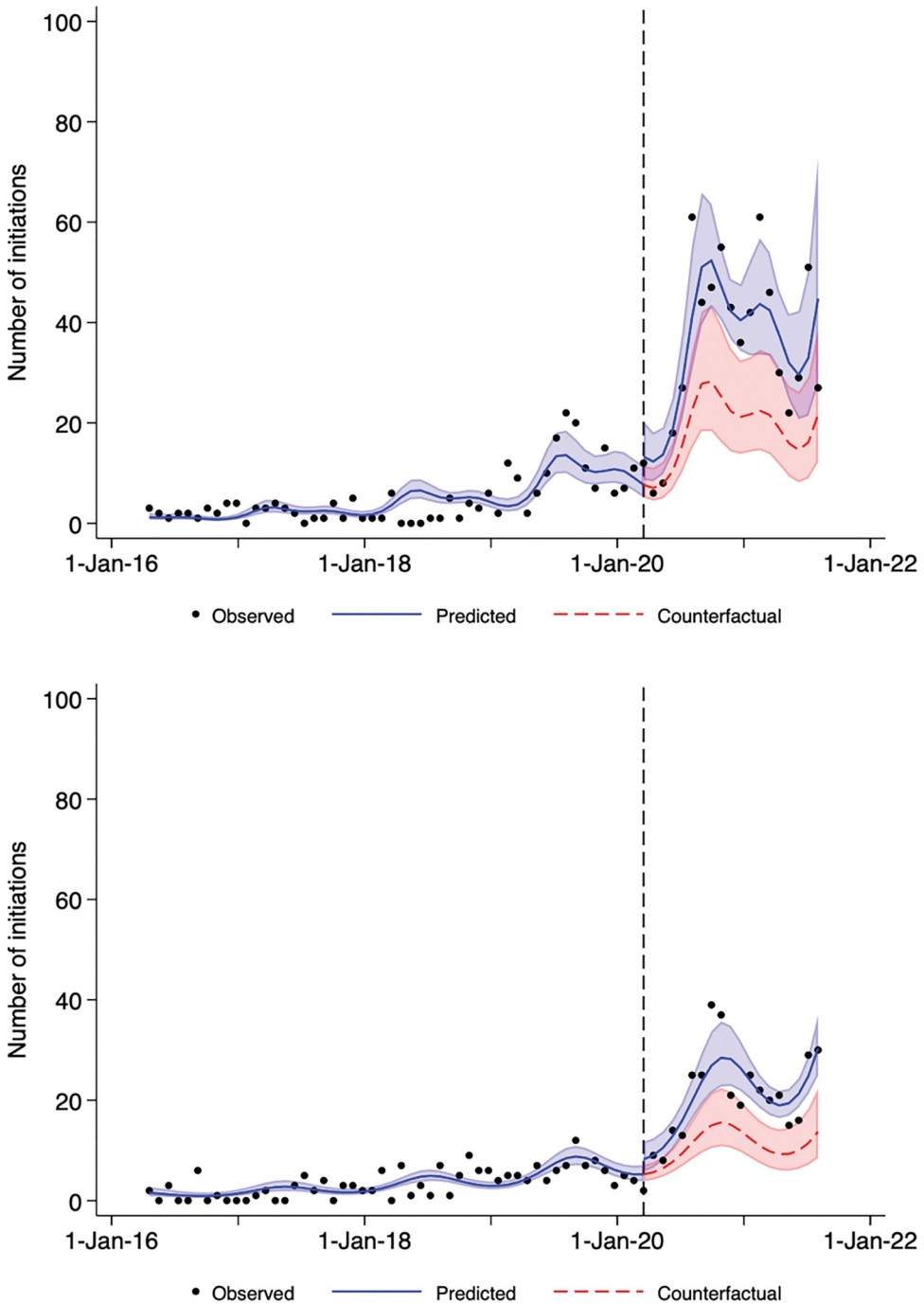

**Fig 2.** New Safer Supply Hydromorphone Dispensations over Time in Ontario for A) male sex and B) female sex. Dots represent the number of safer supply hydromorphone dispensations per 28-day period between early 2016 and late 2021 in Ontario. The vertical dashed line indicates March 17, 2020, the date that Ontario declared an emergency due to the COIVD-19 pandemic. The solid blue line indicates predictions from the best-fitting Poisson model and light blue shading indicates confidence bounds around these estimates. The dashed red line indicates the estimated number of dispensations if the pre-pandemic trends had continued during the pandemic (i.e., the counterfactual scenario) and light red shading indicates confidence bounds around these estimates.

Compared with individuals prescribed SSHM before the pandemic, those prescribed SSHM during the pandemic had a significantly higher prevalence of recent overdose and had more frequently been prescribed opioid agonist therapy in the previous year. These findings suggest that clinicians prescribed SSHM to individuals at high risk of opioid-related toxicity. Overdose rates rose during the pandemic, which may be due to increased toxicity of the unregulated drug supply, physical distancing, and reduced access to services, which may also explain the higher incidence of recent overdose in the pandemic cohort [27]. Our observation that the prevalence of HIV, recent infective complications, and palliative care service use was higher among individuals initiated before the onset of the pandemic may relate to patient selection criteria in defined programs; the first formal SSHM program in London, Ontario, initially prioritized individuals with comorbidities such as untreated HIV and infections that placed them at increased risk of death [28]. Later initiations may reflect the expansion of SSHM outside of formal programs and broadening of inclusion criteria to include those at highest risk of overdose-related death from using the street supply of fentanyl.

While rates of SSHM prescribing increased immediately following the onset of the pandemic, the subsequent trend was similar to that prior to the pandemic, despite rising overdose rates and increasing toxicity of the unregulated drug supply [27, 29], which are known motivating factors for some prescribers to offer SSHM [12]. The lack of a continued increase in rate of prescribing may relate to the ability of SSHM programs to enrol new patients. Despite additional federal funding that may have contributed to the immediate rise seen after the pandemic onset, SSHM programs have substantial waitlists and may have reached capacity as a result of increased demand during the pandemic [12, 30]. While we cannot distinguish SSHM prescribed through an established program from that prescribed by individual clinicians, previous research indicated that a small number of prescribers (likely within established programs) prescribe most SSHM in Ontario [7]. Additionally, our study extended through August 2021, when COVID-19 mortality rates were declining [31]. For prescribers who viewed SSHM as an emergency measure in the context of the pandemic, this changing landscape may have affected their willingness or motivation to prescribe. Ongoing opposition to SSHM prescribing among some practitioners in the addiction medicine community and members of the public may also have influenced the choice not to prescribe SSHM [32].

SSHM has been framed as a strategy to reduce morbidity and mortality among individuals at highest risk of death from overdose and complications related to unregulated drug use [4, 13, 33]. The demographic profile of individuals prescribed SSHM throughout the study period fits with those known to be at highest risk of overdose death in Ontario [9]. We did not find a statistically significant change in the number of females prescribed SSHM during the pandemic, although our study may have been underpowered for this analysis. While the prevalence of opioid use disorder is lower in women, rates of opioid-related harms including overdose are increasing more steeply among women compared to men, and there are important sex and gender-based disparities in access to treatment [34]. Further study is needed regarding sex- and gender-specific access to, and outcomes from, SSHM.

Despite an increase in prescribing due to the pandemic, the overall number of individuals initiated on SSHM during the study period of more than 5 years was less than 1500. This is a low number compared with both conventional opioid agonist therapy and other jurisdictions. There were over 26,000 initiations of opioid agonist therapy in 2021 in Ontario [35]. In British Columbia, the province with the second highest absolute number of overdoses in Canada after Ontario [1], over 4,500 individuals were prescribed SSHM from March 2020 through March 2021 [36]. The higher number of initiations in British Columbia is likely due to release of an official guidance document in support of "Risk Mitigation" prescribing, a COVID-specific variation of SSHM prescribing, that was rapidly released in March 2020 in conjunction with

several leading British Columbia health-related governing bodies [13]. In contrast, SSHM did not receive official support from Ontario's leading addiction medicine or governmental agencies, which may have deterred some prescribers from offering SSHM or led to a lack of awareness of how to implement SSHM. With over 5,200 prescribers of opioid agonist therapy in Ontario in 2021, our study indicates that the vast majority of these practitioners are not prescribing SSHM. Taken together, our findings suggest that a large gap remains in the ability to access SSHM for individuals wishing to do so in Ontario.

## Limitations

Our study has some limitations. First, while our case definition of SSHM has face validity, has been previously published, and is based on consultation with safer supply prescribers and characteristic prescribing patterns [7], it was not assessed for criterion validity. Thus, we may have misclassified some individuals receiving immediate release hydromorphone for an indication other than safer supply. Further, individuals in our study could have received hydromorphone prior to the index date, which may have led to an inaccurate start date. It is reassuring that our results remained consistent in a sensitivity analysis using a more restrictive definition. Conversely, in an attempt to ensure specificity of our definition, we may have excluded individuals who were prescribed SSHM, thereby underestimating the number of initiations. Additionally, our case definition was not designed to include initiations that occurred within recovery and isolation hotels established in Ontario in response to the pandemic, which have particular prescribing characteristics not captured in our current definition. Prescribing of SSHM is known to have occurred within these sites but was frequently discontinued upon discharge [15, 37]. Our study therefore applies only to SSHM prescribed outside of institutional settings. We did not include an offset term in our regression model since we anticipated that the number of people with opioid use disorder in Ontario was relatively stable over the time course of our analysis and accurate assessments of the population of people who use opioids is unavailable. Finally, Ontario's administrative health data does not permit analysis based on gender rather than sex, which is an important consideration that we could not address.

## Conclusions

The onset of the COVID-19 pandemic was associated with an immediate increase in SSHM prescribing in Ontario. Despite an initial expansion in prescribing, the rate of increase of new SSHM initiations was not significantly different during the months after the onset of the pandemic compared with before. Individuals prescribed SSHM during COVID-19 in Ontario continued to be predominantly men at high risk for overdose. While the pandemic seems to have contributed to initiations of SSHM, the overall number of individuals who accessed this intervention remained extremely low compared with opioid agonist therapy and another Canadian jurisdiction. Given the ongoing escalation of the drug toxicity crisis and emerging data signaling benefit experienced by some individuals prescribed SSHM [28], identifying real and perceived barriers to prescribing is an important target of future research.

## Supporting information

**S1 Appendix. Formula for study model equation.**
(DOCX)

**S2 Appendix. Strengthening the Reporting of Observational studies in Epidemiology (STROBE) checklist.**
(DOC)

**S1 Fig. New safer supply hydromorphone dispensations over time in Ontario for sensitivity analysis excluding individuals who received 4mg or 8mg hydromorphone in the 14 days prior to the index date.** Dots represent the number of safer supply hydromorphone dispensations per 28-day period between early 2016 and late 2021 in Ontario. The vertical dashed line indicates March 17, 2020, the date that Ontario declared an emergency due to the COIVD-19 pandemic. The solid blue line indicates predictions from the best-fitting negative binomial regression model and light blue shading indicates confidence bounds around these estimates. The dashed red line indicates the estimated number of dispensations if the pre-pandemic trends had continued during the pandemic (i.e., the counterfactual scenario) and light red shading indicates confidence bounds around these estimates.
(TIFF)

**S2 Fig. New safer supply hydromorphone dispensations over time in Ontario for sensitivity analysis using an alternate intervention date of May 12, 2020.** Dots represent the number of safer supply hydromorphone dispensations per 28-day period between early 2016 and late 2021 in Ontario. The vertical dashed line indicates May 12, 2020, the alternate change date. The solid blue line indicates predictions from the best-fitting negative binomial regression model and light blue shading indicates confidence bounds around these estimates. The dashed red line indicates the estimated number of dispensations if the pre-pandemic trends had continued during the pandemic (i.e., the counterfactual scenario) and light red shading indicates confidence bounds around these estimates.
(TIFF)

**S1 Table. Diagnostic codes used to define aspects of cohort definition and descriptive variables.**
(DOCX)

**S2 Table. Full table of results (including constant and Fourier transformation terms) for primary, sex-specific, and sensitivity analyses comparing initiations of safer supply hydromorphone before and during the COVID-19 pandemic.**
(DOCX)

## Acknowledgments

This study and its data are drawn from the traditional territory and home of many diverse Indigenous people from across Turtle Island. We thank the research team and Community Advisory Committee for the Alterations in Prescribing of Opioids in Response to the Pandemic (ALT-POP) study team for their consultation.

## Author Contributions

**Conceptualization:** Samantha Young, Tara Gomes, Ahmed M. Bayoumi.

**Data curation:** Daniel McCormack.

**Formal analysis:** Samantha Young, Ahmed M. Bayoumi.

**Methodology:** Tara Gomes, Janet Raboud.

**Supervision:** Ahmed M. Bayoumi.

**Writing – original draft:** Samantha Young, Ahmed M. Bayoumi.

**Writing – review & editing:** Tara Gomes, Gillian Kolla, Daniel McCormack, Zoë Dodd, Janet Raboud.

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
