## [Decision Letter · Decision Letter 0]

28 Sep 2023

PONE-D-23-28061Initiations of Safer Supply Hydromorphone Increased During the COVID-19 Pandemic in Ontario: An Interrupted Time Series AnalysisPLOS ONE

Dear Dr. Young,

Thank you for submitting your manuscript to PLOS ONE. After careful consideration, we feel that it has merit but does not fully meet PLOS ONE’s publication criteria as it currently stands. Therefore, we invite you to submit a revised version of the manuscript that addresses the points raised during the review process.

We look forward to receiving your revised manuscript.

Kind regards,

Abdelwahab Omri, Pharm B, Ph.D, Laurentian University

Academic Editor

PLOS ONE

Journal Requirements:

"This study and its data are drawn from the traditional territory and home of many diverse Indigenous people from across Turtle Island. This project was supported by grant funding from the Canadian Institutes of Health Research (CIHR) (grant # 448935) and the Ontario Ministry of Health (grant #709).  This study was supported by ICES, which is funded by an annual grant from the Ontario Ministry of Health (MOH) and the Ministry of Long-Term Care (MLTC). This work was also supported by the Ontario Health Data Platform (OHDP), a Province of Ontario initiative to support Ontario’s ongoing response to COVID-19 and its related impacts. Parts of this material are based on data and information compiled and provided by the Ontario Ministry of Health, Ontario Health (OH) and CIHI. This document used data adapted from the Statistics Canada Postal CodeOM Conversion File, which is based on data licensed from Canada Post Corporation, and/or data adapted from the Ontario Ministry of Health Postal Code Conversion File, which contains data copied under license from ©Canada Post Corporation and Statistics Canada. We thank IQVIA Solutions Canada Inc. for use of their Drug Information File. The analyses, conclusions, opinions and statements expressed herein are solely those of the authors and do not reflect those of the funding or data sources; no endorsement is intended or should be inferred. Samantha Young is supported by the CIHR Vanier Canada Graduate Scholarships and the International Collaborative Addiction Medicine Research Fellowship (NIDA grant R25-DA037756). Gillian Kolla is supported by a Canadian Network on Hepatitis C (CanHepC) Postdoctoral Fellowship and a Canadian Institutes of Health Research Banting Postdoctoral Researcher Award. Tara Gomes is supported by a Tier 2 Canada Research Chair. Ahmed Bayoumi is supported by the Baxter and Alma Ricard Chair in Inner City Health at St. Michael’s Hospital and the University of Toronto. We thank the research team and Community Advisory Committee for the Alterations in Prescribing of Opioids in Response to the Pandemic (ALT-POP) study team for their consultation. "

"This project was supported by grant funding from the Canadian Institutes of Health Research (CIHR; https://cihr-irsc.gc.ca/e/193.html) (grant # 448935) and the Ontario Ministry of Health (https://www.health.gov.on.ca/en/) (grant #709).  This study was supported by ICES (https://www.ices.on.ca/), which is funded by an annual grant from the Ontario Ministry of Health (MOH) and the Ministry of Long-Term Care (MLTC). This work was also supported by the Ontario Health Data Platform (OHDP), a Province of Ontario initiative to support Ontario’s ongoing response to COVID-19 and its related impacts. Parts of this material are based on data and information compiled and provided by the Ontario Ministry of Health, Ontario Health (OH) and CIHI. This document used data adapted from the Statistics Canada Postal CodeOM Conversion File, which is based on data licensed from Canada Post Corporation, and/or data adapted from the Ontario Ministry of Health Postal Code Conversion File, which contains data copied under license from ©Canada Post Corporation and Statistics Canada. We thank IQVIA Solutions Canada Inc. for use of their Drug Information File. The analyses, conclusions, opinions and statements expressed herein are solely those of the authors and do not reflect those of the funding or data sources; no endorsement is intended or should be inferred. Samantha Young is supported by the CIHR Vanier Canada Graduate Scholarships and the International Collaborative Addiction Medicine Research Fellowship (NIDA grant R25-DA037756). Gillian Kolla is supported by a Canadian Network on Hepatitis C (CanHepC) Postdoctoral Fellowship and a Canadian Institutes of Health Research Banting Postdoctoral Researcher Award. Tara Gomes is supported by a Tier 2 Canada Research Chair. Ahmed Bayoumi is supported by the Baxter and Alma Ricard Chair in Inner City Health at St. Michael’s Hospital and the University of Toronto. The funders had no role in study design, data collection and analysis, decision to publish, or preparation of the manuscript."

4. Please upload a copy of Supporting Information Figure/Table/etc. S1 Figure which you refer to in your text on manuscript.

Reviewers' comments:

Reviewer's Responses to Questions

**Comments to the Author**

1. Is the manuscript technically sound, and do the data support the conclusions?

Reviewer #1: Yes

Reviewer #2: Partly

2. Has the statistical analysis been performed appropriately and rigorously? 

Reviewer #1: Yes

Reviewer #2: Yes

3. Have the authors made all data underlying the findings in their manuscript fully available?

Reviewer #1: No

Reviewer #2: Yes

4. Is the manuscript presented in an intelligible fashion and written in standard English?

Reviewer #1: Yes

Reviewer #2: Yes

5. Review Comments to the Author

Reviewer #1: Excellent ITS paper on safer supply hydromorphone use in Ontario before and during the COVID-19 pandemic. Analysis has been conducted well. The SSHM definition has been used previously but has not been compared to a gold standard. This is a significant though not fatal limitation which has been acknowledged and an alternate definition tested. A few minor comments:

1. The primary outcome is the count of new dispensations of SSHM in successive 28-day periods. I understand the interest in new users but do not understand why an individual can only contribute 1 dispensation to the entire study (2016-2020)(line 150). This fact is made more puzzling by sentence before (Line 148), which offers a definition of discontinuation that is not used elsewhere in the eligibility or analysis sections. This makes me think the authors may have intended to allow individuals to contribute another new dispensation to the analysis after they had discontinued SSHM. This seems more appropriate, and aligned with other new user definitions, than excluding an individual from the rest of the analysis after a single dispensation.

2. There are a two references to “new initiations” which is a redundant phrase.

3. The text of the results states that the best fitting model was Poisson, but the caption of the figure states that the models are negative binomial.

4. In table 2, please clarify where the line labelled “Pandemic trend” is truly the trend only within the pandemic period, or rather the change in the trend between the periods. It is interpreted as the latter in the text of the results section.

5. I was unable to see the S1 Figure.

Reviewer #2: Thank you for this paper outlining an interrupted time series analysis to measure the number of new safer supply hydromorphone prescriptions since the COVID-19 pandemic in Ontario. This is an important paper as it addresses a critical public health issue and addresses a novel intervention that has been relatively underexplored in this context. I think publishing this paper would bolster the available scientific literature on this topic. I propose a few minor changes to strengthen the paper before publication.

Abstract

The conclusion is a bit confusing. The phrases ‘abrupt increase’ and ‘rate of increase was similar before’ seems to be contradictory. Perhaps use a different word in place of abrupt or clarify if you mean the number of dispensations.

Introduction

Lines 74-76: You mention that OAT may not meet PWUD’s goals. I wonder if reference should also be made to the fact that many people who are dying from the toxic drug crisis do not have a substance use disorder and, therefore, would likely not benefit from treatment

Are you also including other stimulants when thinking about toxic drug-related deaths? PHAC provides data on both opioids and stimulants, and many deaths are not solely caused by opioids but rather polysubstance use.

Line 90 - maybe be helpful to know if additional programs were added or if only existing programs were expanded. Is there a way of estimating or describing where these programs are located in Ontario in addition to the number of clients that you provided? For instance, are they all located in the GTA? Or are programs spread across Ontario?

Line 93-94 - are there guidelines for Ontario?

Methods

Overall methods are very clear. A few questions to clarify:

Line 132 - why did you choose 2016 as the starting year?

Line 147 - may wish to indicate why those with a cancer diagnosis are excluded. I assume because they would be prescribed pain medication.

Line 167 - may want to provide details as to why seasonality was accounted for

How was sex defined in the data set? Was it self-identified or based on something else? Sex can be measured in a number of ways, including sex of cells, reproductive organs etc.

Results

Line 217 - “have an opioid-related toxicity” is worded awkwardly. Do you mean overdose?

Table 1 - Looking at the sex data, I wonder if there was a significant difference between men and women prescribed SSMH over the two periods in total? If so, this may speak to some of the additional gender-related barriers to accessing harm reduction interventions. If there are data to support this, it could be a valuable contribution.

Discussion

Line 315 - this is a bit confusing because I believe I read earlier in the paper that there was not a significant increase in prescribing between the pre and post pandemic period.

Lines 339-342 - This section may benefit from some conversation about changes to public health measures for COVID-19 during this time (e.g. lockdowns, changes to accessing services) and the impact on this population

Lines 346-349 - what about the federal funds to expand safer supply programs? Were new programs implemented? Are there any efforts to encourage more physicians to prescribe SSHM? You may wish to explore some of these factors. Could ongoing public debate/stigma play a contextual role?

Line 363 - did you calculate power? If so, include in methods.

Lines 371-375 - It may be more effective to compare rates of SSMH prescribing among PWUD rather than crude numbers (as the number of PWUD will differ between provinces)

6. PLOS authors have the option to publish the peer review history of their article (what does this mean?). If published, this will include your full peer review and any attached files.

Reviewer #1: No

Reviewer #2: No

---

## [Author Response · Author response to Decision Letter 0]

9 Nov 2023

Journal Requirements:

We have reviewed our manuscript and file names to ensure they meet PLOS ONE’s style requirements.

"This study and its data are drawn from the traditional territory and home of many diverse Indigenous people from across Turtle Island. This project was supported by grant funding from the Canadian Institutes of Health Research (CIHR) (grant # 448935) and the Ontario Ministry of Health (grant #709). This study was supported by ICES, which is funded by an annual grant from the Ontario Ministry of Health (MOH) and the Ministry of Long-Term Care (MLTC). This work was also supported by the Ontario Health Data Platform (OHDP), a Province of Ontario initiative to support Ontario’s ongoing response to COVID-19 and its related impacts. Parts of this material are based on data and information compiled and provided by the Ontario Ministry of Health, Ontario Health (OH) and CIHI. This document used data adapted from the Statistics Canada Postal CodeOM Conversion File, which is based on data licensed from Canada Post Corporation, and/or data adapted from the Ontario Ministry of Health Postal Code Conversion File, which contains data copied under license from ©Canada Post Corporation and Statistics Canada. We thank IQVIA Solutions Canada Inc. for use of their Drug Information File. The analyses, conclusions, opinions and statements expressed herein are solely those of the authors and do not reflect those of the funding or data sources; no endorsement is intended or should be inferred. Samantha Young is supported by the CIHR Vanier Canada Graduate Scholarships and the International Collaborative Addiction Medicine Research Fellowship (NIDA grant R25-DA037756). Gillian Kolla is supported by a Canadian Network on Hepatitis C (CanHepC) Postdoctoral Fellowship and a Canadian Institutes of Health Research Banting Postdoctoral Researcher Award. Tara Gomes is supported by a Tier 2 Canada Research Chair. Ahmed Bayoumi is supported by the Baxter and Alma Ricard Chair in Inner City Health at St. Michael’s Hospital and the University of Toronto. We thank the research team and Community Advisory Committee for the Alterations in Prescribing of Opioids in Response to the Pandemic (ALT-POP) study team for their consultation. "

"This project was supported by grant funding from the Canadian Institutes of Health Research (CIHR; https://cihr-irsc.gc.ca/e/193.html) (grant # 448935) and the Ontario Ministry of Health (https://www.health.gov.on.ca/en/) (grant #709). This study was supported by ICES (https://www.ices.on.ca/), which is funded by an annual grant from the Ontario Ministry of Health (MOH) and the Ministry of Long-Term Care (MLTC). This work was also supported by the Ontario Health Data Platform (OHDP), a Province of Ontario initiative to support Ontario’s ongoing response to COVID-19 and its related impacts. Parts of this material are based on data and information compiled and provided by the Ontario Ministry of Health, Ontario Health (OH) and CIHI. This document used data adapted from the Statistics Canada Postal CodeOM Conversion File, which is based on data licensed from Canada Post Corporation, and/or data adapted from the Ontario Ministry of Health Postal Code Conversion File, which contains data copied under license from ©Canada Post Corporation and Statistics Canada. We thank IQVIA Solutions Canada Inc. for use of their Drug Information File. The analyses, conclusions, opinions and statements expressed herein are solely those of the authors and do not reflect those of the funding or data sources; no endorsement is intended or should be inferred. Samantha Young is supported by the CIHR Vanier Canada Graduate Scholarships and the International Collaborative Addiction Medicine Research Fellowship (NIDA grant R25-DA037756). Gillian Kolla is supported by a Canadian Network on Hepatitis C (CanHepC) Postdoctoral Fellowship and a Canadian Institutes of Health Research Banting Postdoctoral Researcher Award. Tara Gomes is supported by a Tier 2 Canada Research Chair. Ahmed Bayoumi is supported by the Baxter and Alma Ricard Chair in Inner City Health at St. Michael’s Hospital and the University of Toronto. The funders had no role in study design, data collection and analysis, decision to publish, or preparation of the manuscript."

We have removed all funding information from the Acknowledgements section. We do not have any amendments to our funding statement as written above.

We have included captions for Supporting Information files at the end of the manuscript. 

4. Please upload a copy of Supporting Information Figure/Table/etc. S1 Figure which you refer to in your text on manuscript.

We have uploaded a copy of S1 Figure, which has been split into S1 Figure and S2 Figure according to the journal’s naming preferences.

Reviewers' comments:

5. Review Comments to the Author

REVIEWER #1:

We thank Reviewer 1 for describing our manuscript as an “excellent ITS paper on safer supply hydromorphone use in Ontario before and during the COVID-19 pandemic”. 

1. The primary outcome is the count of new dispensations of SSHM in successive 28-day periods. I understand the interest in new users but do not understand why an individual can only contribute 1 dispensation to the entire study (2016-2020)(line 150). This fact is made more puzzling by sentence before (Line 148), which offers a definition of discontinuation that is not used elsewhere in the eligibility or analysis sections. This makes me think the authors may have intended to allow individuals to contribute another new dispensation to the analysis after they had discontinued SSHM. This seems more appropriate, and aligned with other new user definitions, than excluding an individual from the rest of the analysis after a single dispensation.

The reviewer noted some confusion arising from our choice to include only first dispensations during the study period. While we agree that including more than one dispensation per individual could be informative, our research question focused on initial access to safer supply hydromorphone (SSHM). Including re-initiations of SSHM addresses a different, and more complex, research question that encompasses concerns such as retention in care, stability of patient and provider preferences and relationships over time, competing risks due to incarceration and migration, and other issues. We believe that our focus on first dispensation addresses a well-defined and less confounded research question than including all initiations. Additionally, our previous work (1) has shown that re-initiations of SSHM are rare, with fewer than 18% of initiations being prescribed to individuals with a previous prescription for SSHM. 

We also agree that the statement on discontinuation is not relevant to our analysis and this statement has been removed. 

2. There are a two references to “new initiations” which is a redundant phrase.

All instances of “new initiations” have been changed to “initiations”.

3. The text of the results states that the best fitting model was Poisson, but the caption of the figure states that the models are negative binomial.

We thank the reviewer for noticing this error. We have corrected the captions to reflect the Poisson model used.

4. In table 2, please clarify where the line labelled “Pandemic trend” is truly the trend only within the pandemic period, or rather the change in the trend between the periods. It is interpreted as the latter in the text of the results section.

We have added the additional text at the bottom of Table 2, which we hope has clarified what “pandemic trend” is referring to:

“Pre-pandemic trend refers to the trend from March 22, 2016 to March 16, 2020. Pandemic level change refers to the change at the onset of the pandemic on March 17, 2020. Pandemic trend refers to the trend from March 17, 2020 to August 30, 2021.”

5. I was unable to see the S1 Figure.

We hope that in our revised submission, supplementary figures are now visible. 

REVIEWER #2: 

We thank reviewer 2 for stating that our paper “addresses a novel intervention that has been relatively underexplored in this context” and that “publishing this paper would bolster the available scientific literature on this topic”. We appreciate their helpful suggestions and comments, which we hope have been adequately addressed as follows.

Abstract

The conclusion is a bit confusing. The phrases ‘abrupt increase’ and ‘rate of increase was similar before’ seems to be contradictory. Perhaps use a different word in place of abrupt or clarify if you mean the number of dispensations.

We have changed the word “abrupt” to “immediate”, which we hope better reflects the increase in prescribing that occurred at the onset of the pandemic (rather than the ensuing rate of prescribing that occurred longitudinally after the pandemic). 

Introduction

Lines 74-76: You mention that OAT may not meet PWUD’s goals. I wonder if reference should also be made to the fact that many people who are dying from the toxic drug crisis do not have a substance use disorder and, therefore, would likely not benefit from treatment

We agree with the reviewer that many people who are dying from the toxic drug crisis do not have a diagnosable substance use disorder and therefore OAT is not typically indicated or prescribed. However, SSHM in the current prescribed model in Ontario is generally only offered and prescribed to individuals who meet criteria for an opioid use disorder. Therefore, we feel it may be confusing to raise this point since we are not describing a prescribing model that would address the issue of toxic drug deaths among individuals who do not meet criteria for opioid use disorder. 

Are you also including other stimulants when thinking about toxic drug-related deaths? PHAC provides data on both opioids and stimulants, and many deaths are not solely caused by opioids but rather polysubstance use.

The reviewer raises an important point about the drug toxicity crisis extending beyond opioids to include stimulants and other substances. It is for this reason that throughout the manuscript we use the term “drug toxicity crisis” instead of “opioid crisis” to reflect the potential contribution of stimulant and polysubstance use to overdose deaths. 

Line 90 - maybe be helpful to know if additional programs were added or if only existing programs were expanded. Is there a way of estimating or describing where these programs are located in Ontario in addition to the number of clients that you provided? For instance, are they all located in the GTA? Or are programs spread across Ontario?

We have added further information in line 91 that the funding mentioned expanded safer supply programs in Toronto, London, Hamilton and Ottawa, Ontario. A published environmental scan of safer supply programs in Canada provides further context around the location of safer supply programs in Canada and is referenced in our manuscript (2). It is important to note that our analysis includes SSHM prescribing both within established programs as well as outside of these programs, as we discuss in lines 376-378.

Line 93-94 - are there guidelines for Ontario?

We added further information on the Ontario guidelines (line 96):

“In Ontario, the pre-existing safer supply guidance document was updated for the pandemic context (3).”

Methods

We appreciate the reviewer saying that overall methods are very clear”

Line 132 - why did you choose 2016 as the starting year?

We chose 2016 as the starting year since that was when the first SSHM program is known to have started in Ontario. We have now explicitly stated this in the manuscript on line 136:

“We chose to begin the study period in 2016 because it is when SSHM prescribing first occurred in Ontario.”

Line 147 - may wish to indicate why those with a cancer diagnosis are excluded. I assume because they would be prescribed pain medication.

We have added the following text to explain why individuals with a cancer diagnosis were excluded:

“We also excluded individuals who had a cancer diagnosis or cancer-related treatment in the 1 year before the index date (S1 Table) because of the potential for hydromorphone prescribed for pain to be misclassified as SSHM.”

Line 167 - may want to provide details as to why seasonality was accounted for

We have added additional information to explain why seasonality was accounted for (line 172):

“First, we assessed the model for seasonality after visual inspection of trends over time suggested periodic trends in the data.”

How was sex defined in the data set? Was it self-identified or based on something else? Sex can be measured in a number of ways, including sex of cells, reproductive organs etc.

We have expanded upon the definition of sex in the administrative database in line 164:

“In addition to analyzing new dispensations overall, we performed subgroup analyses by male and female sex (based on sex documented at birth).”

Results

Line 217 - “have an opioid-related toxicity” is worded awkwardly. Do you mean overdose?

We hope we have clarified the wording as follows (line 223) as well as specified that the definition for opioid-related overdose is found in S1 Table:

“and were more likely to have had an opioid-related overdose in the prior 1 year (29.6% versus 14.1%, p=<0.001) (S1 Table)”

Table 1 - Looking at the sex data, I wonder if there was a significant difference between men and women prescribed SSMH over the two periods in total? If so, this may speak to some of the additional gender-related barriers to accessing harm reduction interventions. If there are data to support this, it could be a valuable contribution.

We agree that sex is an important consideration in our analysis as well as in the broader discussion around SSHM prescribing. We did not assess differences between men and women using statistical hypotheses as these were exploratory secondary analyses that should be viewed as hypothesis-generating rather than hypothesis-testing. Furthermore, the best-fitting model was slightly different for both men and women, making a direct statistical comparison challenging. 

As the reviewer notes, we found a lower absolute number of female sex individuals prescribed SSHM compared to male sex individuals. The number of initiations over the two periods in total are reported for both sexes in the section titled “sex-specific analyses” (line 273). We discussed some of the complexities around interpreting this data (line 392):

“While the prevalence of opioid use disorder is lower in women, rates of opioid-related harms including overdose are increasing more steeply among women compared to men, and there are important sex and gender-based disparities in access to treatment (4). Further study is needed regarding sex- and gender-specific access to, and outcomes from, SSHM.” 

We also note a limitation of our analysis is that we are only able to analyze the data by sex and not gender (line 435).

Discussion

Line 315 - this is a bit confusing because I believe I read earlier in the paper that there was not a significant increase in prescribing between the pre and post pandemic period.

Thank you for letting us know that our summary of the results was confusing. We have added some text to this section (now line 332) as follows:

“We found that rates of prescribing of SSHM were increasing prior to the pandemic. Prescribing increased significantly and immediately at the time that the COVID-19 public health emergency was declared. The rates of prescribing during the pandemic were explained by a combination of both the pre-pandemic trend and the immediate increase. That is, we did not observe an additional increase in prescribing rates during the pandemic beyond this combination.”

Lines 339-342 - This section may benefit from some conversation about changes to public health measures for COVID-19 during this time (e.g. lockdowns, changes to accessing services) and the impact on this population

We have added further discussion about pandemic-related changes and the effects on this population (line 357):

“Overdose rates rose during the pandemic, which may be due to increased toxicity of the unregulated drug supply, physical distancing, and reduced access to services (5).”

Lines 346-349 - what about the federal funds to expand safer supply programs? Were new programs implemented? Are there any efforts to encourage more physicians to prescribe SSHM? You may wish to explore some of these factors. Could ongoing public debate/stigma play a contextual role?

The reviewer raises numerous important additional considerations that may have influenced practitioners’ decision to prescribe safer supply (or not) and help contextualize our results. We have highlighted several of them with additional text:

Line 373 – “Despite additional federal funding that may have contributed to the immediate rise seen after the pandemic onset, SSHM programs have substantial waitlists and may have reached capacity as a result of increased demand during the pandemic”

Line 382 – “Ongoing opposition to SSHM prescribing among some practitioners in the addiction medicine community and members of the public may also have influenced the choice not to prescribe SSHM (6).”

Line 363 - did you calculate power? If so, include in methods.

As a population-based study, our analysis included all available data (that is, we did not select a sample of the population for analysis). As such, we did not perform power calculations prior to our analysis. We believe that providing confidence intervals will allow readers to assess the precision of our estimates. Notably, the confidence intervals for non-significant parameters in Table 2 are all narrow.

Lines 371-375 - It may be more effective to compare rates of SSMH prescribing among PWUD rather than crude numbers (as the number of PWUD will differ between provinces)

We agree with the reviewer that a rate of SSHM prescriptions out of the population of PWUD would be more informative than an absolute number. However, there are no readily available estimates of the populations of PWUD in each province during this time frame to allow for this comparison. We do include the following statement to help contextualize the fact that the number of overdose deaths are higher in Ontario compared to BC (line 401):

“In British Columbia, the province with the second highest absolute number of overdoses in Canada after Ontario…”

REFERENCES:

1. Young S., Kolla G., McCormack D., Campbell T., Leece P., Strike C. et al. Characterizing safer supply prescribing of immediate release hydromorphone for individuals with opioid use disorder across Ontario, Canada, Int J Drug Policy 2022: 102: 103601.

2. Glegg S., McCrae K., Kolla G., Touesnard N., Turnbull J., Brothers T. D. et al. "COVID just kind of opened a can of whoop-ass": The rapid growth of safer supply prescribing during the pandemic documented through an environmental scan of addiction and harm reduction services in Canada, Int J Drug Policy 2022: 106: 103742.

3. Hales J., Kolla G., Man T., O’Reilly E., Rai N., Sereda A. Safer Opioid Supply Programs (SOS): A Harm Reduction Informed Guiding Document for Primary Care Teams; 2020.

4. Barbosa-Leiker C., Campbell A. N. C., McHugh R. K., Guille C., Greenfield S. F. Opioid Use Disorder in Women and the Implications for Treatment, Psychiatr Res Clin Pract 2021: 3: 3-11.

5. Gomes T., Murray R., Kolla G., Leece P., Bansal S., Besharah J. et al. Changing circumstances surrounding opioid-related deaths in Ontario during the COVID-19 pandemic, Toronto, Ontario: Ontario Drug Policy Research Network, Office of the Chief Coroner for Ontario and Ontario Agency for Health Protection and Promotion (Public Health Ontario); 2021.

6. Lam V. As a doctor, I was taught ‘first do no harm.’ That’s why I have concerns with the so-called ‘safe supply’ of drugs: The Globe and Mail; 2021.

---

## [Editor Report · Decision Letter 1]

15 Nov 2023

Initiations of Safer Supply Hydromorphone Increased During the COVID-19 Pandemic in Ontario: An Interrupted Time Series Analysis

PONE-D-23-28061R1

Dear Dr. Samantha Young,

We’re pleased to inform you that your manuscript has been judged scientifically suitable for publication and will be formally accepted for publication once it meets all outstanding technical requirements.

Kind regards,

Abdelwahab Omri, Pharm B, Ph.D, Laurentian University

Academic Editor

PLOS ONE

---

## [Editor Report · Acceptance letter]

20 Nov 2023

PONE-D-23-28061R1 

Initiations of Safer Supply Hydromorphone Increased During the COVID-19 Pandemic in Ontario: An Interrupted Time Series Analysis 

Dear Dr. Young:

I'm pleased to inform you that your manuscript has been deemed suitable for publication in PLOS ONE. Congratulations! Your manuscript is now with our production department. 

Kind regards, 

on behalf of

Dr. Abdelwahab Omri 

Academic Editor

PLOS ONE